# Dietary Sources of Salt in Low- and Middle-Income Countries: A Systematic Literature Review

**DOI:** 10.3390/ijerph16122082

**Published:** 2019-06-12

**Authors:** Elias Menyanu, Joanna Russell, Karen Charlton

**Affiliations:** 1School of Medicine, Faculty of Science, Medicine and Health, University of Wollongong, Northfields Avenue, Wollongong, NSW 2522, Australia; ekm963@uowmail.edu.au; 2School of Health and Society, Faculty of Social Sciences, University of Wollongong, Northfields Avenue, Wollongong, NSW 2522, Australia; jrussell@uow.edu.au; 3Illawarra Health and Medical Research Institute, Wollongong, NSW 2522, Australia

**Keywords:** dietary salt, sources of salt, salt intake in LMICs, systematic literature review

## Abstract

Rapid urbanization in low- and middle-income countries (LMICs) is transforming dietary patterns from reliance on traditional staples to increased consumption of energy-dense foods high in saturated fats, trans fats, sugars, and salt. A systematic literature review was conducted to determine major food sources of salt in LMICs that could be targeted in strategies to lower population salt intake. Articles were sourced using Medline, Web of Science, Scopus, and grey literature. Inclusion criteria were: reported dietary intake of Na/salt using dietary assessment methods and food composition tables and/or laboratory analysis of salt content of specific foods in populations in countries defined as low or middle income (LMIC) according to World Bank criteria. Of the 3207 records retrieved, 15 studies conducted in 12 LMICs from diverse geographical regions met the eligibility criteria. The major sources of dietary salt were breads, meat and meat products, bakery products, instant noodles, salted preserved foods, milk and dairy products, and condiments. Identification of foods that contribute to salt intake in LMICs allows for development of multi-faceted approaches to salt reduction that include consumer education, accompanied by product reformulation.

## 1. Introduction

Cardiovascular diseases (CVDs) are the leading cause of death worldwide [1], with hypertension accounting for more than 50% of premature deaths [2]. Globally, the prevalence of hypertension has been rising from a figure of 25% reported in 2000 [3] to 40% reported by others in 2008 [4]. Hypertension is the leading single risk factor contributing to overall Global Burden of Disease [2] through its association with cardiovascular disease. Low- and middle-income countries (LMICs) already share the highest prevalence of hypertension [5], with predictions that three quarters of the world’s hypertensive population will be found in these countries within the next 10 years [3]. This may be in part due to the larger population sizes in LMICs compared to high-income countries, but also because of the inability of their health care systems to cope with the management of chronic diseases. This results in large numbers of people with undiagnosed, untreated, and uncontrolled hypertension [6]. 

Low- and middle-income countries are currently facing an unprecedented hypertension burden [6]. The past four decades have seen a shift in the highest blood pressure (BP) levels from high-income countries to LMICs, particularly in some South Asian and sub-Saharan African countries [7] where more than a third of adults report being hypertensive [6]. Additionally, there is low awareness, treatment, and control measures for hypertension in LMICs [8], rendering current practices to reduce hypertension ineffective [9,10]. Ultimately, the worsening situation of hypertension together with increased prevalence of cancers, diabetes, and chronic respiratory diseases have culminated in nearly 80% of all deaths from non-communicable diseases (NCDs) occurring in LMICs [11].

The nutrition transition in many LMICs is occurring due to the rapid urbanization which has led to widespread availability of energy-dense foods that are high in saturated fats, trans fats, sugars, and salt. This has resulted in a shift in dietary patterns from a reliance on traditional staples such as maize and sorghum to more processed foods [12]. 

At the same time, food insecurity remains a challenge in LMICs [13]. Preserving food using traditional methods (e.g., salting) is important in ensuring food availability and addressing hunger in many communities [14,15]. Further, limited water supplies [16] have exposed populations to unhealthy water sources which often contain toxins or unacceptable levels of nutrients [17,18]. A mean Na concentration of about 700 mg/L (with extremes exceeding 1500 mg/L) was found in drinking water in coastal areas of Bangladesh [19] contributing to the overall daily Na consumption of people living along the coast. With this, the WHO’s recommended daily limit of 5 g salt [20] can easily be exceeded by just drinking 2–3 L of water in some countries.

There is compelling evidence from epidemiological, clinical, and experimental studies showing a positive and direct relationship between salt consumption and BP. It is widely accepted that high intakes of salt in food (beyond the WHO’s recommended level of 5 g salt or 2 g Na/day) and in water (>0.2 g Na/L) are major risk factors for hypertension [20,21], heart disease, and stroke [22]. Sodium (Na) is an important nutrient required by the body to ensure acid-base balance, maintenance of plasma volume, and transmission of nerve impulses [23]; however, when in excess, has been implicated in the development of kidney disease, gastric cancer, and hypertension [24,25]. Lower levels of Na (<3 g/d) have also been identified to be associated with higher risk of death and cardiovascular events [26]. Several prospective cohort studies have indicated a U-shaped relationship between salt consumption and cardiovascular disease or mortality, with increased risk at both high- and low-intake extremes [27,28,29,30]. In comparison with a moderate consumption of salt, observational data demonstrate that very high intakes (>6 g Na/day, representing only 10% of the population studied) are associated with an excessive risk of cardiovascular events and death, but only in the case of those with hypertension. This study also reported associations between low salt intakes and increased risk of cardiovascular events and death in both hypertensives and normotensives [31]. While the optimal lowest intake of salt is still being debated, there is no question that in most countries, population-level estimates of salt far exceed the recommendation of a maximum 5 g/day, thus, necessitating salt reduction strategies [20].

In 2004, the WHO released its “Global Strategy on Diet, Physical Activity and Health” that was adopted by the World Health Assembly and which called upon all governments and stakeholders to work towards improving the healthfulness of diets [32]. More recently, WHO member states have agreed to work towards voluntary targets to reduce NCDs by 25% by 2025, with one of the nine global targets being a reduction in population salt intake by a relative 30% [22].

In order to develop national policies and strategies to lower population-level salt intake in LMICs, it is necessary to have an understanding of the main dietary sources of salt. This information is required to assist governments to enact policies and programs, and to concentrate on particular food items that are of relative importance to excessive salt consumption in LMICs. The aim of this systematic literature review is to identify the major food sources that contribute to salt consumption in populations in LMICs.

## 2. Materials and Methods 

A literature search was conducted in March 2017 using the Medline, Web of Science, and Scopus databases. Hand searching of the reference lists was performed from articles retrieved from these databases. Grey literature was also searched by visiting institution and government websites and other sources. Data collected were synthesized according to the Preferred Reporting Items for Systematic Reviews and Meta-Analyses (PRISMA) [33] (Figure 1). The research question investigated was: “What are the dietary sources of salt in LMICs?” Search terms are listed in Appendix A. Outcome measures included dietary sources of Na/salt. Only articles reported in English were included. All countries in the LMIC bracket were listed among the search terms. Low-income economies were defined as those with a gross national income (GNI) per capita, calculated using the World Bank Atlas method, of $995 or less in 2017; lower middle-income economies were those with a GNI per capita between $996 and $3895; and upper middle-income economies were those with a GNI per capita between $3896 and $12,055 [34]. The search was restricted to articles published from 1960 onwards. 

Data extraction was completed by the lead author (EM) and reviewed by the second author (KC). Endnote X4 (Clarivate Analytics, Philadelphia, PA, USA) was used to manage the citations. Each article was ranked for level of evidence using the National Health and Medical Research Council (NHMRC) recommendations [35]. A narrative synthesis of the studies was completed because of the heterogeneity in reporting outcomes of the studies. In this literature review, Na and salt were used interchangeably. Studies were included if a study (a) reported actual quantities of Na/salt in foods and (b) data were collected from food diaries, diet recalls, food frequency tables or laboratory analysis of salt content in food, and (c) laboratory analysis of salt content in food.

Data were extracted by first reviewing the title and selecting abstracts of those with relevant titles. If the abstract met the inclusion criteria, the full article was retrieved (see Figure 1). Studies were summarized according to descriptive characteristics, including the region, country, data collection period, study emphasis, and the study design. Each paper was categorized also by the author, population, outcome measured, method of measurement, and the results. Quality rating was conducted using appraisal tools for cross-sectional studies for nine articles that utilized a human population [36] (Appendix A). This systematic review was registered with the International Prospective Register of Systematic Reviews (Prospero CRD42016038173).

## 3. Results

Our search of databases and additional records yielded 3207 titles. One-hundred-and-seventy-seven abstracts (23 hand-searched) were retrieved after duplicates and irrelevant papers were excluded, thereafter resulting in 34 full papers being assessed for eligibility and inclusion. Of these 34 articles, 15 were included in the final review. Reports from the country level WHO Stepwise Surveillance surveys were hand-searched but did not yield relevant information. Figure 1 shows the PRISMA flow diagram providing details of the search and included studies.

Of the final included articles grouped according to WHO-defined geographical regions [37], the greatest number of articles (i.e., four each) were from Africa and the Western Pacific (though in the Western Pacific region, all the articles came from China), whereas the least number of the articles (i.e., one) came from European countries (Appendix A). Fifteen of the included studies were conducted in 12 countries and most of them had been published within the last two decades (1991–2016). Six of the articles did not report data collection periods. With the exception of one randomized controlled trial, all other studies were cross-sectional and descriptive in design and concentrated on dietary behavior and product reformulation.

Analytical values for the salt content of bread were reported in five articles (Table 1). Zibaeenezhad et al. [38] included the highest number of samples of bread that were examined, whereas Ferrante et al. [39] recorded the greatest number of bakeries that were sampled within one study. For studies that included samples of bread from different suburbs and locations, salt content varied depending on the municipality and producer. The highest salt content found in bread was 1.80 g/100 g in Nigeria [40]. The average salt content in bread from all articles was more than the voluntary targets and recommended quantities in other countries (380–400 mg Na per 100 g bread) [41,42,43]. One study from Ferrante et al. [39] analyzed salt content in French bread, croissants, and cookies and crackers using dietary recall and additionally used biochemical analysis for French bread. Results from the two methods used for French bread indicated an underestimation of salt use with the dietary recall. Financial constraints did not permit chemical analysis of the other food items.

There were three articles that reported salt content of additional foods other than bread, including preserved fish, cookies, crackers, as well as water (Table 2) and seven articles assessed total dietary salt intake (Table 3). Two of the included articles [44,45] investigated children’s’ dietary salt intake. Four of the articles that measured dietary intake also included biomarkers of participants’ 24 h urinary Na excretion [46,47,48,49]. The major sources of dietary salt were breads, meat and meat products, bakery products, noodles, salted preserved foods, milk and dairy products, and condiments. Additionally, some articles reported that a high amount of salt in the local diet came from discretionary salt [45,49,50], but this was not consistently measured.
ijerph-16-02082-t001_Table 1Table 1Summary table of studies that reported salt content of bread by chemical analysis.ReferencePopulationOutcome Measured Method of MeasurementResultsSilva et al., 2015 [51]All bakeries (*n* = 17) situated in Maputo city that were listed in the Mozambican Yellow Pages were included. Na content in bread.Flame photometryMean Na content of bread was 450 mg/100 g, ranging between 255 mg/100 g and 638 mg/100 g, with no significant differences between bakeries and traditional markets. Most samples (88%) did not meet the regulation for South Africa.Mean Na > a, b, and c.Nwanguma and Okorie, 2013 [40]Retail samples of 100 brands of white bread made from wheat flour, representing the major brands were purchased from 10 standard retail outlets in Nsukka and Enugu towns, both in Enugu State in South-Eastern Nigeria. Na content in bread.Flame photometryMean Na = 544 mg/100 g. Na ranged from 396 mg to 1332 mg/100 g.Mean Na > a, b, c, and d. Hussain and Takruri, 2016 [52] 68 samples of seven types of bread were collected from 13 different bakeries in the city of Amman, Jordan.Na content in breadFlame photometryMean Na content = 476 ± 84 g/100g ranging between 168 ± 20 g for * White Arabic bread to 824 ± 76 g/100 g for * shrak bread.Mean Na > a, b, and c.Zibaeenezhad et al., 2010 [38]204 bakeries in districts of Shiraz city in Iran; 408 bread samples were collected from bakeries, measuring the salt content of 6 different kinds of bread.Na content in bread.Laboratory testing of salt percentage in bread as outlined by Iran’s Organization for Standards and Industrial Investigations [53]Mean Na = 524 g/100 g ranging from 0–1400 g/100 g bread.Mean Na > a, b, c, and d.Vukić et al., 2013 [54]12 samples of bread purchased in stores from the 3 municipalities: Bijeljina, Zvornik, and East Sarajevo in Bosnia. In each municipality 8 samples were randomly selected. Na content in bread.Atomic absorption spectrophotometry(AAS) using an instrument VARIAN Spectr AA-10 [55]Mean Na = 405 ± 177 mg/100 g, 489 ± 174 mg/100 g, and 673 ± 119 mg/100 g for East Sarajevo, Bijeljina, and Zvornik, respectively.Bread samples from East Sarajevo, mean Na > a and b.Bread samples from East Bijeljina, mean Na > a, b, and c.Bread samples from East Zvornik, mean Na > a, b, c, d, and e.Ferrante et al., 2011 [39]25,000 bakeries countrywide affiliated to Argentinean Federation of Bakeries. Na content in bakery products.Dietary recall and flame photometry Self-reported (using food composition table) mean Na content of French bread = 1.8% (range 1.0% to 4.0%), chemical analysis of French bread, mean Na concentration = 2.0% (range 1.4% to 3.0%) of total salt intake.Mean Na > a, b, c, d, and e* White Arabic and shrak bread are bread types in Amman, 2% Na ≈ 4 g of total salt intake. a > 380 mg/100 g—maximum level of Na in bread established by the South African Government; effective June 2019 [41]. b > 400 mg/100 g—maximum level of Na in bread recommended by the Government of Australia [42,43]. c > 450 mg/100 g—maximum level of Na in bread recommended by the National Heart Foundation of New Zealand [42]. d > 490 mg/100 g—Level of Na that is required by the Finnish Government for the designation of “highly salty” on a label [56]. e > 550 mg/100 g—maximum level of Na in bread established by the Portuguese Government [57].

## 4. Discussion

To reduce population salt intake, the WHO “SHAKE the Salt” framework strongly emphasizes a need to identify major sources of salt in the diet and to embark on strategies that include product reformulation, taxation, nutrition labelling, and nutrition education [69]. The current review has identified bread as the major food source that provided the highest amounts of salt to the diets of populations in 12 LMICs. Meat and meat products, salted meat and fish, sauces, spreads, condiments, pizza, sandwiches, seafood, and ground or river water were also identified as major contributors to non-discretionary salt intake. Contribution of each of these foods differs by cuisine of the country, for example among Asian countries, sauces and MSG are prevalent [45,50,66], whereas in Africa and Latin America it is bakery, meat, and dairy products [40,44,49].

The average Na content for bread was above 400 mg per 100 g bread, which exceeded the generally accepted target for salt content of bread [41,42,43,70]. Similarly, in high-income countries such as Australia, some loaves provide more than 25% of the maximum daily recommended intake of salt in bread in just two slices [71], with bread and bread rolls contributing 25% of total salt provided by processed foods [72]. The consumption of bread has increased in LMICs [73] due to the decline in traditional staples such as maize, millet, and sorghum to the rise in wheat [74]. In India, the production of wheat increased from 75.81 million MT in 2007 to 94.88 million MT in 2012 [75], while Pakistan experienced a 3.2% increase between 2012 and 2013 [76]. Food preferences have changed and this has altered the consumption of traditional staples. Relatively, there has been a global shift in dietary patterns from the reliance on traditional staples to processed foods due to the increased production of processed foods and changing lifestyles [20]. The Food and Agriculture Organization predicts that with the rise in income levels, preference for wheat products will increasingly overshadow traditional coarse grains [77]. In many LMICs, wheat is gradually replacing roots and tubers [78]. Furthermore, rapid urbanization in LMICs is changing diets such that “fast foods” (ready-to-eat), which typically contain excess salt, have replaced core staple plant-based foods, thereby increasing the demand for processed foods [79]. Other reasons for shifts to a greater reliance on processed foods may be related to an increase in the number of women in the workforce, which has affected roles within the household related to food preparation [80]. The rising trend of eating away from home or eating prepared food away from home has contributed to the increased consumption of bakery products [81,82] in urban settings. Bread has virtually become a staple food [83], and which for many, forms the basis of breakfast, lunch, and dinner. For example, in some parts of the Democratic Republic Congo, bread has substituted cassava, the traditional starchy staple used in preparing “foufou” (dough made from boiled and ground plantain or cassava) [84]. Similarly, bread intake is higher than traditional staples in southern Mozambique [85,86] and is widely eaten as a snack or as a complete meal in Nigeria [87]. Bread is easily accessible from stores and vendors and more convenient than traditional cereal and root staples, both of which require time and effort for preparation before consumption. In the Seychelles, there has been a decreased consumption of the traditional staple (fish) as a result of increased intake of meat, poultry, processed meat, and snacks [88].

There have been major increases in meat (i.e., beef, pork, and poultry) in LMICs [89,90], including processed meat that is high in salt [91,92]. Increased consumption of processed meat products has the potential risk of coronary heart disease, type-2 diabetes, and colorectal cancer [93,94]. In Australia, processed meat contributed 10% of daily Na intake [95] and is described as being a major contributor to salt intake in the United Kingdom [96]. Bacon for example, contains more than twenty times the quantity of salt compared to fresh pork of the same weight [97]. As countries are becoming more economically wealthy, there has been an increased demand for animal source products with livestock being one of the fastest growing agricultural subsectors in LMICs [98], and accompanying advancement in food technology in these countries resulting in increased availability of processed animal products. These processed animal products may be contributing to the increase in salt intake within LMIC populations. A potential strategy to reduce salt intake would be to reduce processed meat intake. This would have the additional benefit of reducing climate change issues such as high greenhouse gas emissions from cattle and diverting agricultural land use from human consumption to animal consumption [99,100].

Highly palatable, energy-dense, low-cost, ultra-processed foods, snacks, and beverages are commonplace within the food supply in LMICs [101,102]. This is evident in the growing number of supermarkets that are replacing open public markets in LMICs [12]. Consumers are attracted to these foods because of their affordability, ease of access, availability, and taste, which are often accompanied by intensive marketing by the food industry sector [103], particularly in poorer environments where alternatives are limited [104]. 

In most LMICs, food security is a major issue and food preservation using salt is commonplace. In these countries, food production is seasonal but in greater quantities during harvest, after which production dwindles significantly [14,15]. Ghana, for example, experiences two rainy seasons a year (with the major one in June and the minor in October) and those are the times for crop production in most parts of the country [105]. The major fishing season shares the same timeframe [106,107], thus, the bulk of the country’s food supply is produced in one-half of the year, thereby warranting preservation, one of which is using salt [108] to ensure its availability and maintenance for off-season supply [14,15]. 

An often under-estimated source of dietary salt relates to the contribution from the water supply. Studies on river and underground shallow water salinity are lacking in the face of severe drought in LMICs [109,110,111]. More challenging is the fact that there are no guidelines for safe salinity levels in drinking water [21]. During the dry season in Bangladesh, it is estimated that up to 5–16 g salt per day may be consumed from river and shallow underground drinking water [47]. There are many places in Africa and Asia where the sources of water are unsafe for human consumption [112,113,114] and the likelihood for excess salt provided by water bodies remain high as environmental and climatic conditions worsen [115]. This issue has been ignored in recommended global strategies [20] to reduce population-level salt intake. 

The shift in dietary patterns is also driven by the promotion of Western culture through media outlets, international trade, and other channels related to globalization [74]. Many supermarkets, hotels, restaurants, and fast food outlets in LMICs are multinational establishments that provide the same menu as their partners in higher income countries. These food outlets are a major driving force changing the food environment and dictating food preferences in LMICs [12]. Globalization of food systems in this way, undermines traditional dietary practices and creates an avenue for food and nutrient insecurity. Greater availability and access to these cheap, imported, energy-dense, nutrient-poor foods have culminated in an increased prevalence of obesity and a myriad of NCDs plaguing many African countries (that are already burdened with infectious diseases) [116]. A challenge is for LMICs to consider agriculture within economic growth and development, to ensure that food insecurity is not exacerbated by increasing urbanization. A deliberative approach should be developed and geared towards the production, availability, accessibility, and consumption of a wide range of traditional staples in the face of rapid development. Countries in South East Asia provide exemplars as they have encouraged and improved the traditional cuisine to meet current developmental changes [117,118]. The United Nations’ Sustainable Development Goals provide a framework against which country progress towards 17 goals will be reported by governments [119]. Partnerships and global solidarity between countries should include commitments by major food companies to limit and standardize acceptable maximum limits of harmful components allowed in imported or branded foods sold across the globe and prevent dramatic differences currently seen in the salt content of similar products and fast food meals across countries.

An urgent need to address food systems in LMICS is required in order to stem the pandemic of NCDs, as more than three-quarters of premature NCD deaths occur in these countries [20]. South Africa serves as an example where the government has mandated maximum salt levels permitted across a wide range of processed foods, in an effort to reduce the burden of hypertension [8,120,121] and its related morbidities [122]. Though the impact of this legislation is yet to be demonstrated [123], it is seen as a bold step towards saving lives and reducing health care costs [124,125]. The WHO has emphasized that government policies and programs, collaboration with private sector organizations, and monitoring of population salt intake are key measures to population salt reduction [20].

Understanding salt use behaviors at the population level is required in order to determine best approaches to salt reduction in countries [126]. Many governments have focused on individual consumer behavior change [127], focusing on salt added at the table and during cooking. Though commendable, a multi-faceted approach which targets both discretionary and non-discretionally salt use is encouraged [20]. This review collated the available information on dietary sources of salt in LMICs in an effort to identify the food sources that contribute to total salt intake. Many LMICs do not have reliable assessments of dietary salt intake nor do they have national targets and timelines towards ensuring a decline in salt consumption. This makes it challenging to assess progress towards achieving the WHO’s voluntary salt reduction target of a relative 30% by 2025 [20]. Advocacy by civil society groups, organizations, institutions, and professional health associations for salt reduction in foods is required to lobby governments whose health agenda does not include changes to the food supply. This approach has been demonstrated by the Australian Division of World Action on Salt and Health, whose efforts have resulted in voluntary salt targets for the food industry [71]. The legislative approach adopted by the South African government [41] may serve as blueprint for LMICs [128].

Limitations to the review relate to the selection criteria being based on the English language only. This may have limited the scope of the search and subsequent data retrieved for consideration. Nonetheless, studies included covered all WHO global regions. Many of the cited papers did not provide information on the actual dietary consumption patterns of the populations residing in the areas from where the food samples were obtained. This means that overall contribution of individual foods to total salt intake could not be determined. 

## 5. Conclusions

Processed foods that have a high salt content are becoming commonplace in the diets of populations in LMICs. Major sources of salt are provided by bread, meat and meat products, salted meat and fish, sauces, spreads, condiments, pizza, and seafood. Differences exist between WHO regions regarding sources of food products that contribute to total salt intakes. Water sources in drought-affected areas also warrant further attention as sources of Na intake. Besides foods that contain high salt, discretionary salt use is widespread in LMICs.

## Figures and Tables

**Figure 1 ijerph-16-02082-f001:**
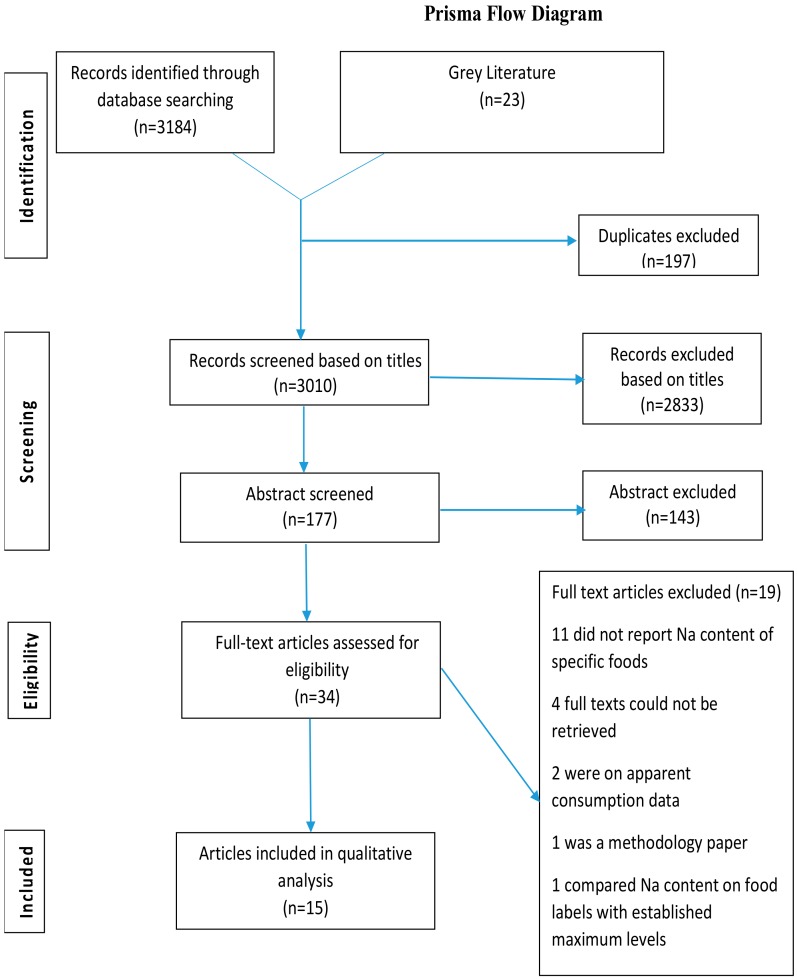
PRISMA 2009 flow diagram.

**Table 2 ijerph-16-02082-t002:** Summary table of studies that assessed consumption of salt from specific foods and water.

Reference	Population	Outcome Measured	Method of Measurement	Results
Kerry et al., 2005 [46]	12 villages (6 rural, 6 semi-urban) were chosen in Ghana. Between 95 and 250 subjects aged 40–75 years from each village, for a total 1896, selected by stratified random sampling from a census of all inhabitants of the village.	Frequency of consumption of high salt foods.	Food frequency questionnaire asked about the consumption of five salty foods: koobi, momoni, kako (all salted fish), salted pig’s feet, and salted beef. Also questioned the use discretionary salt, stock cubes or monosodium glutamate (MSG).	92% reported eating salted fish. While salted meat (pig’s feet and beef) was eaten more often by semi-urban villagers, salted fish was eaten more often by rural villagers. Majority of the respondents (98%) frequently added salt to food in cooking.
Ferrante et al., 2011 [39]	25,000 bakeries countrywide affiliated to Argentinean Federation of Bakeries.	Na content in bakery products.	Dietary recall and Flame photometry.	Self-reported (using food composition table) mean Na contents: croissants and cookies = 1.8% (range 1.0% to 3.5%), crackers = 2.9% (range 2.2% to 5.0%), and flat, rounded crackers = 2.1% (range 1.2% to 3.4%) of the total salt intake.
Khan et al., 2011 [47]	343 pregnant Dacope women from Bangladesh recruited for a pilot phase of a larger study.	Na intake from drinking water sources	Indirect estimates of individual salinity intake from groundwater and river water, determined using salinity data for 1998–2000; Centre for Environment and Geographic Information System (CEGIS) in Bangladesh.	Na from drinking water = 2064 mg/day during the dry season (depending on the water source) and 480 mg/day during the monsoon season assuming a conservative water intake of 2 L/day/person.

**Note:** 5 g/d salt = 2000 mg Na, 2% Na ≈ 4 g of total salt intake.

**Table 3 ijerph-16-02082-t003:** Summary table of studies that assessed total salt intake using dietary assessment methods.

Reference	Population	Outcome Measured	Method of Measurement	Results
Charlton et al., 2005 [49]	300 men and women from three different ethnic groups (black, mixed ancestry, and white), aged 20 to 65 y, conveniently sampled from place of work, Cape Town City Council, South Africa. Equal numbers of hypertensive (BP ≥ 140/90 mm Hg and/or on antihypertensive medication) and normotensive (BP < 140/90 mm Hg) men and women were planned (*n* = 150/group, 50 from each ethnic group).	Dietary intake of Na.	Interviewer administered 3 repeated 24 h recalls. Standard household measuring utensils, rulers, and food photographs of typical South African foods [58] used to quantify food portion sizes. The average daily nutrient intake calculated using Foodfinder III computerized dietary assessment program, based on Medical Research Council Food Composition Tables [59].	In all three subsamples, cereals were the main contributor to total reported dietary Na intake (45.9% to 48.6%), followed by meat and meat products (20.3% to 23.6%) and milk and dairy products (6.3% to 8.1%). In all groups, bread was the major source of dietary Na (25.2% to 40.5%).
Liu et al., 2014 [48]	726 Chinese post-menopausal women who attended a screening visit for a randomized controlled trial testing the effect of soy products supplementation on BP were conveniently sampled.	Dietary intake of food substances from which Na content was determined.	A 3-day food records questionnaire was used to estimate dietary nutrients intake. Food items were those most frequently consumed based on previous local surveys [60,61]. Subjects received a 30 min training on estimation of food amounts, portion, and utensil sizes. Dietary nutrients were calculated based on the China Food Composition Table and local Na database [62,63]. Total Na intake was calculated by summing the estimates from all contributory food items or groups.	Major sources of non-discretionary salt include soup (21.6%), rice and noodles (13.5%), baked cereals (12.3%), salted/pre-served foods (10.8%), Chinese dim sum (10.2%), and sea foods (10.1%) of the total salt intake.
Zhao et al., 2015 [45]	903 families were conveniently sampled for the study. 2952 participants were recruited from families in urban (Xicheng District) and suburban (Huairou District) Beijing, China. Study families were recruited through public primary and junior high schools. Eligible families were those with a child from the enrolled schools.	Dietary salt intake and sources of salt in the diet.	Questionnaire; a simplified “one-week salt estimation method” was designed to measure each family member’s daily salt intake and determine the sources of salt in the diet. This method estimates salt intake from three sources: household cooking, processed food, and cafeterias or restaurants. The methodology was previously published [64].	Soy sauce, vinegar, other sauces and MSG contributed 47%, 34%, 12%, and 7% to total Na intake. The mean Na intake was 5360 (SD 3320) mg/day. Adults consumed more Na 6080 (SD 3640) mg/day than children and adolescents 4400 (SD 2480) mg/day and senior citizens 4080 (SD 1920) mg/d.
Health Promotion Board, Singapore, 2011 [65]	Singaporean National Nutrition Survey 2010, comprised 739 subjects aged 18–69 years conveniently sampled.	Na content in selected foods.	Face-to-face interviews were conducted where dietary practices and food frequency questionnaires were administered. Nutrients and various food groups were assessed by comparing the levels of intake with dietary standards including the Recommended Dietary Allowances (RDAs).	Fish balls, fish cakes, breads, and noodles were estimated to contribute 37% of the population’s salt intake. Daily Na intake was 3265 mg/day.
Du et al., 2014 [66]	Secondary data from China Health and Nutrition Survey (1991–2009) comprising 16,869 adults aged 20–60 y were used.	Na intake from foods and condiments.	Three consecutive 24 h dietary recalls in combination with weighing methods. All foods and condiments recorded and measured. Na intake (i.e., Na from all foods and condiments) were based on their compositions in the Chinese food-composition table.	The average soy sauce intake was 6.9 g/d, accounting for 8.5% of total Na intake. The average processed food intake was 244.7 g/d, which represented 20.8% of all food consumed and accounted for 6.8% of total Na intake. The average MSG intake was 1.5 g/d, accounting for 3.4% of total Na intake.
de Moura Souza, 2013 [44]	Nationwide dietary survey. Food consumption of a representative sample of the Brazilian population 10 years of age or older (*n* = 34,003).	Na content in foods and beverages.	24 h dietary recall using the nutrition data system for Research software version 2008, the Brazilian Food Composition Table [67], and Brazilian studies on regional foods [68].	Foods with high Na densities (>600 mg/100 g) included salty preserved meats (997 mg/100 g), processed meats (974 mg/100 g), cheeses (883 mg/100 g), crackers (832 mg/100 g), sandwiches (800 mg/100 g), pizza (729 mg/100 g), and breads (646 mg/100 g), as well as oils, spreads, sauces, and condiments (804 mg/100 g). Altogether these food groups contributed 25% (811 mg/100 g) of the average daily Na intake. The mean Na intake was 3190 mg/day.
Anderson et al., 2010 [50]	Participants were 4680 women and men aged 40 to 59 years, recruited by stratified random sampling from 17 diverse populations—community-based or workplace-based—in Japan (four samples), People’s Republic of China (three rural samples), the United Kingdom (two samples), the United States (eight samples).	Na intake was calculated by summing estimates from all contributory food sources, including foods and beverages, ingested at home or away from home.	24 h dietary recall. Na content of each food item was determined using the enhanced national food database for each country.	For China, mean Na intake = 3990 ± 1943 mg/person/day; Soy sauce = 256 mg/person/day, mustard, turnip greens, and cabbage = 143 mg/person/day, sodium bicarbonate and sodium carbonate (tenderizers) = 98 mg/person/day, and noodles were 89 mg/person/day. Japan, mean Na intake = 4651 ± 1279 mg/person/day, United Kingdom mean Na intake = 3406 ± 1162mg/person/day, and United States mean Na = 3660 ± 343.1 mg/person/day

**Note:** 5 g/d salt = 2000 mg Na, 2%% Na ≈ 4 g of total salt intake.

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
