# Peer review of "Dietary Sources of Salt in Low- and Middle-Income Countries: A Systematic Literature Review"

_ijerph, 2019, doi:10.3390/ijerph16122082_

Round 1

Reviewer 1 Report

The authors present new information on sodium sources in LIMCs that are targets for health policies and interventions. The discussion is thorough, covering many characteristics of LIMCs impacting food and specifically sodium consumption that are likely unknown by readers.

My primary recommendation is to describe in more than one sentence (lines 68-69) the evidence showing adverse health effects of reducing sodium below moderate intake levels. More recent references are suggested below. This suggestion does not detract from the general recommendation to keep sodium intake in check in LIMCs, but rather to remind readers that evidence does not support a recommendation to reduce sodium intake to the lowest level in all people.

Mente A, O’Donnell M, Rangarajan S et al. - The Lancet 2018; 392: 496–506

BMJ. 2019 Mar 13;364:l772. doi: 10.1136/bmj.l772.

Int J Hypertens. 2018; 2018: 6956078.

Other comments:

Reference 38, Ferrante: Article not available to me; is the number 25,000 bakeries correct in Table 1?

Line 64: check sentence structure.

Line 65: use abbreviation, Na, on first use of 'sodium' in previous paragraph.

Author Response

My primary recommendation is to describe in more than one sentence   (lines 68-69) the evidence showing adverse health effects of reducing sodium   below moderate intake levels. More recent references are suggested below.   This suggestion does not detract from the general recommendation to keep   sodium intake in check in LIMCs, but rather to remind readers that evidence   does not support a recommendation to reduce sodium intake to the lowest level   in all people.

Thank you. This has been added in the manuscript (line 67-75).

Reference 38, Ferrante: Article not available   to me; is the number 25,000 bakeries correct in Table 1?

Yes, it is correct.

Line 64: check sentence structure.

Thank you. The sentence has been rephrased   (lines 64, 65).

Line 65: use abbreviation, Na, on first use of   'sodium' in previous paragraph

 Thank   you. The correction has been made and cross checked throughout the   manuscript.

Reviewer 2 Report

The question that is addresses in this study is relevant. However, the discussion lacks a reflection on strengths and limitations of the study. I would recommend to include this.

Some suggestions:

Table 1 shows salt content of bread in different countries. Although it is interesting to see the high quantities of salt in bread, this is only relevant when intake of bread is known. If people consume bread only rarely, it may not contribute significantly to sodium intakes and/or hypertension. Could the authors provide information about bread intake (or indicate when this information does not exist) and about the reasons for showing these sodium contents of bread (and not other food products)?

When looking at Table 2 it is clear that there are differences between regions in food products that contribute to high sodium intakes. It may be fair to mention this in the conclusion as well.

Including publications in English only may affect outcome. There may be more information at local / country level on food intakes of populations. Although it is understandable to include only English publications, it would be helpful to discuss effects of this inclusion criterion in more detail.

Author Response

Thank you for these helpful suggestions.

There is very little data on quantities and frequency of bread consumed in LMICs. We have rather cited  studies that indicate the fact that bread is gradually replacing traditional stables, particularly with rising production of wheat and to the decline of sorghum, maize and millet (this was mentioned in lines 180-184). These studies highlight the increasing consumption of bread in many countries. Although bread contains a relatively moderate amount of salt, its frequency of consumption makes it one of the most important sources of dietary salt (Quilez & Salas-Salvado, 2012). We have included additional information on these occurrences, where in Mozambique, those living in the south of the country are consuming more bread than traditional staples and in Nigeria where bread is widely eaten as a complete everyday meal (lines 198-99).

Aside from bread, we have discussed other food products (tables 2 and 3).

‘Differences between regions in food products that contributed to high sodium intakes’ were highlighted in the discussion (lines 173-175).  We have included a statement to that effect in the conclusion (line 294-295).

We have included study limitations (lines 285-289).

Reviewer 3 Report

This review targets a key topic in nutrition and public health – identification of main approaches in reducing salt intake in real life, focusing into Low and Middle-Income Countries (LMIC). While some of these are already burdened with infectious diseases, the supply of highly processed foods, usually high in salt/sugar/fat is becoming another challenge to cope with. The review has been done systematically according to PRISMA recommendations. Nevertheless, I suggest a few minor modifications:

L43: Some sentences are very strong and should be revised, to have a little less “passion”. Example of such sentence is “The African region for example, shared the greatest burden with more than a third of adults reported being hypertensive”, while in L32 authors are mentioning lobal average of 40%. This would mean that African region is actually having bellow average burden.

L32: Please check if change from 25% to 40% is indeed appropriate for period of 8 years. I seriously doubt that, and if provided numbers are exact, there is probably a major methodological difference between both studies that reported this data. Revise appropriately.

L82; typo, “using” used twice

L137: “recommended quantities” need to be specified as a number, as it is done latter on. Without that this sentence is not appropriate.

L139: did authors mean chemical (and not biochemical) analyses?

Table 1, study No. 4 (Zibaeenezhad et al., 2010): Please extract method of measurement from references in same way as in other provided studies (for example flame photometry or other method provided there)

Table 2, Page 9: Bellow the table there is only “Note. 5g/d salt = 2000mg.” Mentioning “Na” is probably missing.

Table 2, Health Promotion Board, Singapore, 2011 [64] (Page 11): Are you sure that 3265mg/day was provided as salt and not Na intake? This is very low intake for salt.

L195: word “consumption” is probably missing in this sentence

Author Response

L43: Some sentences are very strong and should be revised, to have   a little less “passion”. Example of such sentence is “The African region for   example, shared the greatest burden with more than a third of adults reported   being hypertensive”, while in L32 authors are mentioning lobal average of   40%. This would mean that African region is actually having bellow average   burden.

Thankyou. The statement has been   rephrased (lines 43-44).

L32: Please check if change from 25%   to 40% is indeed appropriate for period of 8 years. I seriously doubt that,   and if provided numbers are exact, there is probably a major methodological   difference between both studies that reported this data. Revise   appropriately.

Thanks for pointing this out. The   has sentence has been rephrased (lines 31-33).

L82; typo, “using” used twice

Thank you. This has been corrected   (line 90).

L137: “recommended quantities” need   to be specified as a number, as it is done latter on. Without that this   sentence is not appropriate.

Thank you. The specified number has   been included (line 142).

L139: did authors mean chemical (and   not biochemical) analyses?

We meant biochemical analysis.

Table 1, study No. 4 (Zibaeenezhad   et al., 2010): Please extract method of measurement from references in same   way as in other provided studies (for example flame photometry or other   method provided there).

Thank you for this suggestion. The   method of measurement is very lengthy. Writing all of it into the table might   make the table look awkward. We thought highlighting and providing a   reference for the method would assist readers in understanding the various   stages of the method.

Table 2, Page 9: Below the table   there is only “Note. 5g/d salt = 2000mg.” Mentioning “Na” is probably missing

Na has now been inserted (lines 162,   165).

Table 2, Health Promotion Board,   Singapore, 2011 [64] (Page 11): Are you sure that 3265mg/day was provided as   salt and not Na intake? This is very low intake for salt.

Thank you for pointing this out.  This was indeed a typo and has been   corrected (Table 3).

L195: word “consumption” is probably   missing in this sentence

Thank you. The omission has been   rectified (lines 199-201).